# Biomechanical Examination of Wrist Flexors and Extensors with Biodex System Dynamometer—Isometric, Isokinetic and Isotonic Protocol Options

**DOI:** 10.3390/medicina60071184

**Published:** 2024-07-22

**Authors:** Marta Jokiel, Katarzyna Kazmierczak, Piotr Czarnecki, Aleksandra Bartkowiak-Graczyk, Anna Madziewicz, Ewa Breborowicz, Malgorzata Miedzyblocka, Michal Adamski, Krystian Kaczmarek, Leszek Kaczmarek, Leszek Romanowski

**Affiliations:** 1Traumatology, Orthopedics and Hand Surgery Department, Poznan University of Medical Sciences, 61-701 Poznan, Poland; 2Physiotherapy Department, Poznan University of Medical Sciences, 61-701 Poznan, Poland; 3Department of Rehabilitation, Poznan University of Medical Sciences, 61-701 Poznan, Poland; 4Traumatology, Orthopedics and Hand Surgery Student Scientific Group, Poznan University of Medical Sciences, 61-701 Poznan, Poland

**Keywords:** biodex, hand examination, wrist flexors, wrist extensors

## Abstract

*Background and Objectives:* Biodex System^®^ is an advanced dynamometer used for testing various biomechanical parameters of muscles. Test outcomes allow for the identification of muscle pathology and consequently lead to a clinical diagnosis. Despite being widely used for the testing and rehabilitation of the human musculoskeletal system, no universal and acceptable protocol for wrist examination has been proposed for patients with wrist pathology. In this study, the authors aim to identify the most appropriate protocol for testing the biomechanical parameters of flexors and extensors of the wrist. *Materials and Methods:* A group of 20 patients with symptomatic tennis elbow and 26 healthy volunteers were examined using three different protocols: isokinetic, isometric and isotonic. Protocol order for each study participant was assigned at random with a minimum of a 24 h break between protocols. All protocol parameters were set according to data obtained from a literature review and an earlier pilot study. Following completion of each protocol, participants filled out a questionnaire-based protocol, assessing pain intensity during the exam, difficulty with exam performance and post-exam muscle fatigue. *Results:* The isotonic protocol showed the best patient tolerance and the highest questionnaire score. There was a significant difference (*p* < 0.05) between the three protocols in average pain intensity reported by study participants. All participants completed the isotonic protocol, but not all patients with symptomatic tennis elbow were able to complete the isometric and isokinetic protocols. The isotonic protocol was deemed “difficult but possible to complete” by study participants. *Conclusions:* The isotonic protocol is most suitable for testing the flexors and extensors of the wrist. It gives the most biomechanical data of all protocols, is well tolerated by patients and rarely causes pain during examination even in symptomatic participants.

## 1. Introduction

A biomechanical exam of joint and muscle function using static and dynamic tests is among the most objective methods of assessing patients and athletes. Biomechanical parameters can be tested by three main protocols which are based on different aspects of physiological muscle contraction. The isokinetic protocol is performed with constant angular velocity which is maintained by machine resistance and movement inhibition. The isotonic protocol is performed with constant muscle tension, and the isometric protocol is performed without movement but with tension shift. Depending on the examination purpose, recommendations for different dynamometers may differ and so may the recommended protocols. For example, the isokinetic protocol is widely recommended for athletes and healthy subjects [1]. Different studies give conflicting information regarding best protocol selection for specific groups of muscles, different patient groups and different clinical conditions [2,3,4,5,6,7,8]. Biodex System 4 Pro^®^ (Biodex Medical Systems, Inc., Shirley, NY, USA) is a highly developed dynamometer that allows the examination of various biomechanical parameters of muscles. Its design enables the study of limb muscles and the vertebral column as well as their movements. Initially, it was used mostly for the examination of athletes, but over the last 15 years, it has become widely used for the biomechanical evaluation of patients before and after orthopedic treatment and physical therapy procedures [9]. Biodex System is widely used for patient evaluation with several medical conditions [10,11,12]. It is a reliable tool for muscle assessment [13]. Patient results are presented in a tabular format, which compares contralateral limbs (involved vs. uninvolved or dominant vs. nondominant) allowing for the detection of differences between them. One of the dynamometer advantages is its ability to provide patients with exercise schemes as well as endurance protocols with biofeedback options [4,14,15]. During biofeedback exercise, the patient has the opportunity to observe the change in biomechanical parameters, such as torque changes in time, which can be modified with specific patient-performed tasks [16]. This seems to be crucial, especially for patients with muscle contraction issues [17,18,19].

This study may be beneficial for muscle biomechanical parameter measurement, especially interconnected with muscle tendinopathies which highly influence patient mechanical abilities and decrease quality of life. One of the top examples is patients with lateral epicondyle enthesopathy—tennis elbow (TE). Knowledge about wrist extensor biomechanical parameter function and potential decrease may be beneficial for patient treatment because it will provide objective measurement and modification of patients’ rehabilitation programs.

Despite the fact that the number of papers regarding Biodex medical usage is growing, only a few researchers tried to examine the biomechanical parameters of upper limb and forearm muscles [20,21]. Unfortunately, neither the manufacturer nor the researchers specify appropriate protocols for patients with pain symptoms.

During wrist examination, it is important to find the proper starting position and ensure there are no co-movements of the elbow, shoulder and trunk [22]. The Biodex System ensures the same patient position during each trial, preventing co-movements in untested joints. This is crucial for biomechanical testing because using additional muscle groups may distort patient results.

Nevertheless, the primary goal is to find the appropriate protocol that will allow for muscular assessment for healthy subjects, as well as patients with forearm pathology such as lateral elbow tendinopathy (TE), wrist instability, etc. [22].

The aim of this study was to find the most appropriate protocol for wrist examination, which can be used to evaluate the biomechanical parameters of wrist flexors and extensors on the Biodex dynamometer.

## 2. Materials and Methods

### 2.1. Materials

The study was approved by the Ethics Committee of Poznan University of Medical Sciences (number 392/15) and was conducted in accordance with the Declaration of Helsinki at the Traumatology, Orthopedics and Hand Surgery Department of the Poznan University of Medical Sciences from 2015 to 2016. All study participants consented in writing to participate in the study prior to its commencement. A detailed description of tests used in the study was provided to the participants who had the right to withdraw from the study at any time without penalty.

Two groups of subjects were selected to participate in the study: patients with painful wrist motion and healthy controls. In order to maintain group homogeneity, symptomatic participants were selected from patients with unilateral lateral elbow tendinopathy (TE), who were diagnosed with the condition by two independent, experienced clinicians based on history and clinical symptoms. Prior to study commencement, the authors completed a pilot study in order to determine most appropriate protocol parameters. The pilot study (PS) participants comprised unilateral tennis elbow patients (*n* = 6) and healthy controls (*n* = 4). Lateral epicondylar pain during wrist movement is the most characteristic and consistent symptom of TE. The pain is predominantly attributed to the ECRB (Extensor carpi radialis brevis) muscle contraction during straightening and extending the hand and wrist and can be assessed by biomechanical examination [23]. The pain declines the quality of life and hinders daily activities such as eating and lifting objects as well as performing manual labor [24]. It therefore becomes crucially important to have the tools to assess patients’ biomechanical status prior to and following TE treatment and to provide a customized exercise program for ECRB muscle strengthening, using, for example, device biofeedback [25].

The main study included 20 patients with symptomatic unilateral TE (13 female, 7 male) and 26 healthy controls (14 female, 12 male). The study participant demographic data are presented in Table 1.

Exclusion criteria for Biodex wrist testing included previous wrist trauma or history of wrist complaints, previous upper limb trauma, peripheral nerve entrapment symptoms and previous surgical or physical therapy treatment of TE.

### 2.2. Methods

During the examination, the patient was seated in the dynamometer chair with the back supported and the shoulders stabilized with vertical belts. The belt’s position eliminated the unwanted co-movements of the shoulder and trunk. The examined limb was positioned with the shoulder adducted at 30 degrees and the elbow flexed to 90 degrees. The forearm was supported by the dynamometer plate in its proximal half. The distant half of the forearm was free to allow for full range of wrist motion. Prior to examination commencement, the patient was instructed to firmly grip a handle, which was attached to the dynamometer. The axis of the dynamometer was aligned with the axis of the wrist joint. During the examination, the patient was instructed to move the handle using only the wrist and refrain from moving the upper part of the body (Figure 1).

The examination position and protocol parameters were assessed by a previous pilot study which was conducted by the authors in the same department.

### 2.3. Pilot Study

According to available literature, to objectively assess participants’ biomechanical parameters, three different trials had to be conducted: warm-up (with light load), strength (with near-maximal resistance to assess muscles’ highest potential) and endurance (with moderate load) [26,27]. In order to determine appropriate protocol parameters, the authors conducted the isometric examination followed by the isotonic and the isokinetic exams. The pilot study began with isometric protocol to assess the forearm muscles peak torque; according to peak torque, we determined the isotonic protocol limitations. After isotonic measurement, we determined the constant speed velocity for isokinetic protocol.

Isometric wrist angle range was chosen based on range most useful in performing daily activities: 30 degrees of extension, 30 degrees of flexion and a neutral position. Each trial consisted of three repetitions of maximal isometric wrist contraction in both flexion and extension [28]. Based on isometric test results, the isotonic resistance was set at 20%, 50% and 70% of isometric peak torque [1,29]. Since only healthy participants were able to complete the isotonic protocol, the constant tension was reduced to 0.5/1/0.5 Nm, which allowed the completion of this protocol also by symptomatic patients. Based on speed values obtained in isotonic testing, the isokinetic test was set with constant speed values of 210/90/150 deg/s. Each protocol consisted of three cycles of wrist flexion and extension. Protocol specifications are presented in Table 2.

After signing formal consent, each participant was randomly assigned to one of the three protocols. After completing this protocol, the participant was again randomly assigned to another protocol. There was a minimum of a 24 h break between protocols [8]. There was a 15 s break between the three trials constituting each of the protocols.

Following completion of each protocol, participants were asked to fill out a questionnaire-based protocol assessing pain intensity in VAS during the exam, individual protocol difficulty assessment in VAS with exam performance and post-exam muscle fatigue (also measured in VAS). Furthermore, study participants were asked to select one protocol that they found most suitable between 3 examined protocols.

### 2.4. Data Analysis

All data were analyzed with the Statistica13.3^®^ program (TIBCO^®^ at Poznan University of Medical Sciences license, Fredry 10, 61-701 Poznan, Poland). Using the Chi-squared test, it was determined that a sample consisting of at least 37 subjects was needed to obtain 80% power with α = 0.05 type I error and β = 0.20 type II error. G*Power analysis provides total sample size of 45 within 0.95 power measured with G.Power 3.1^®^ software shared by Heinrich Heine Dusseldorf University. The variables were investigated using visual (histograms, probability plots) and statistical methods (Kolmogorov–Smirnov test) to determine whether or not they were normally distributed. Descriptive analyses were presented using mean and standard deviation (SD) for the normally distributed variables and median and interquartile range (IQR) for the non-normally distributed and ordinal variables. For normally distributed variables, TE and CG groups were compared with the *t*-test, while the three protocols were compared with the one-way ANOVA test. For non-normally distributed variables, TE and CG groups were compared with the Mann–Whitney U test, while the protocols were compared with the Kruskal–Wallis test. Association between protocols was assessed with the Tukey (for one-way ANOVA) and Dunn (for Kruskal–Wallis) post hoc analysis. The significance level for all tests was set at *p* <  0.05.

## 3. Results

### 3.1. Pain

Based on subjective participant assessment using VAS, the isotonic protocol examination caused the least amount of pain in both the TE and the CG groups, while the isometric protocol examination was reported as the most painful by study participants (Table 3). Moreover, participants in the TE group reported a significant difference (*p* < 0.05) in pain intensity during examination between the isotonic protocol and the other two protocols.

### 3.2. Difficulty

Based on subjective participant assessment using VAS, the isotonic protocol examination caused the least difficulty in both the TE and the CG groups, while the isometric protocol examination was reported as the most difficult by study participants (Table 4). Moreover, participants in the TE group reported a significant difference (*p* < 0.05) in the examination difficulty level between the isotonic protocol and the other two protocols.

### 3.3. Fatigue

Based on subjective participant assessment using VAS, the isotonic protocol examination caused the least fatigue upon completion in both the TE and the CG groups, while the isometric protocol examination was reported as causing the greatest fatigue by study participants (Table 5). Moreover, participants in both TE and CG groups reported a significant difference (*p* < 0.05) in the post-examination fatigue level between the isotonic protocol and the other two protocols.

In the TE group, three participants did not complete the isokinetic protocol examination, while ten participants did not complete the isometric protocol examination due to excessive pain. All TE group participants completed the isotonic protocol. In the CG group, all participants completed all three protocol examinations. Most of the TE participants and majority of the CG participants chose the isotonic protocol as the most comfortable and confirmed that they would retake the examination (Table 6).

The isotonic and isokinetic protocols can be completed in approximately 5 min each, while the isometric protocol requires over 15 min to complete.

All protocols give constant biofeedback to participants and allow for the adjustment of muscle performance during the examination. Due to the assessed protocols, the isotonic protocol may be considered accurate for patient wrist extensor and flexor biomechanical dynamic measurement with pain issues and muscle strength decrease. The isotonic protocol may be considered accurate for patient wrist extensor and flexor biomechanical dynamic measurement when the patient is able to perform higher restriction activities and among sportsmen. The isometric protocol may be considered accurate for patient wrist extensor and flexor biomechanical static measurement.

The authors’ summary of the advantages and limitations of Biodex wrist examination protocols is presented in Table 7.

The isotonic protocol yields the most information regarding the biomechanical parameters of muscles, by measuring and recording 20 different data points, whereas the isokinetic and isometric protocols measure 11 and 8 data points, respectively. The particular parameters measured in each of the three study protocols are listed in Table 8. The quantity of biomechanical parameters allows for an accurate assessment of muscle work quality.

## 4. Discussion

Our study reveals that the isotonic protocol is the most suitable for the biomechanical examination of wrist flexors and extensors. The Biodex System protocols are reliable and validated techniques that give a unique opportunity to measure different types of muscle function [5]. This type of dynamometer is frequently used as a training device that can be easily employed for musculoskeletal examination [6,20,30].

It is generally acknowledged that clinicians use isometric and isotonic contraction to build and assess muscle mass and isokinetic training as a method for improving muscular functional performance [31].

Our results correspond with a study by Roy et al. who focused on finding the most appropriate biomechanical protocol for shoulder girdle examination. Their results showed that the isotonic protocol is best for assessing shoulder girdle muscle fatigue, especially the rotator cuff muscles. Roy et al. also stressed the short time required to complete the isotonic protocol examination (less than 10 min) compared to the isometric examination [1]. Other authors reported that during the knee joint exam on the Biodex System, the isotonic work activates more motor units than the isokinetic work and that the isotonic work is most related to the physiological activity of muscles [32,33]. The aforementioned results also correspond with study outcomes by Driessche et al. who focused on the biomechanical evaluation of muscle aging. As a result of this study, the isotonic measures are functionally relevant methods for muscle aging detection [34]. The isotonic protocol may be sufficient for muscle assessment not only for patients in pain but also for patients with additional muscle weakness. In our study, we focused on patients with muscle pain while performing functional tasks with resistance, but according to the literature, it provides more possibilities among different groups of patients.

The isotonic protocol allows the measurement of the work fatigue parameter which is crucial in the examination of athletes. Biomechanical data concerned with symmetrical or asymmetrical muscle work may be a useful predictor of muscle overload and overuse in various sports [35].

The isokinetic protocol is widely used in Biodex System dynamometer studies [3,26,36,37,38]. However, most of these studies examine athletes or healthy subjects. Ellenbecker et al., who conducted a study with young tennis players, reported that the isokinetic protocol was difficult for the study participants and caused pain during testing. We can presume that patients in pain may have problems with finishing the isokinetic protocol [39]. Similar problems were reported by Kaymak et al. who showed that during the wrist isokinetic test, the wrist position was crucial and the protocol was difficult for participants [40].

For the isokinetic protocol, the subjects performed movement with the maximal angular velocity of 210 deg/s, 90 deg/s and 150 deg/s. The first two velocities were chosen mainly because the power output for each angular velocity is expected to be different based on classic Newtonian mechanics [41]. Alternatively, other authors examining forearm muscle strength on the Biodex System have chosen velocities of 60 and 180 [29]. We did not select these proposed values because our pilot study demonstrated that the resistance generated by the machine was too large, and even healthy test participants had difficulties with task completion. The manufacturer recommendations were 60 and 120 deg/s for wrist flexor and extensor examination, respectively, and it required much effort for healthy young participants from the pilot study to finish the task. In the pilot study, it was impossible for symptomatic patients to complete the exam using the manufacturer’s recommended values. For this reason, we decided to change the recommended parameters adjusting them to patients’ ability.

For the isometric protocol, we used angular positions: 30 degrees of dorsal flexion, 30 degrees of palmar flexion and a neutral position. This angular range enables patients to perform most of the activities of daily living and is attainable by both healthy participants and patients with symptomatic TE [42,43].

For the isotonic protocol, we used constant values of tension 0.5/1/0.5 Nm. We did not use greater tension than 1 Nm, because following the pilot study, we observed that participants were not able to complete the protocol with higher tension values. We repeated the tension of 0.5 Nm in the first and the last trials since only the isotonic protocol can measure work muscle fatigue. Despite experiencing pain during the examination, the patients with symptomatic TE were able to complete the isotonic protocol. We can presume that it will be useful in diagnosing patients with many different wrist pathologies or patients following wrist surgery. Nagle et al. and Folchert et al. indicate that it is extremely important to find a wrist measurement assessment tool, which can be widely used not only for gaging treatment outcomes but also for biofeedback use [44,45]. Forearm muscle examination with the isotonic protocol may be a useful tool for the evaluation and treatment of different clinical conditions of the wrist such as wrist instability and wrist pain [46,47,48].

Albanese GA et al. indicate that it is difficult to define a framework for patients with forearm muscle issues that will be presented as the dynamic fatigue-related changes associated with wrist function. In their work, they used the superficial electromyography protocol which enables the measurement of muscle fatigue but does not reveal specific muscle biomechanical parameters such as muscle torque, velocity or muscle fatigue. This is why the isotonic protocol may be considered as the standard tool for measuring patient muscle fatigue during active performance [49].

Hill CA et al., during their examination of patients with TE, also indicated that there are biomechanical limitations for objective patient assessment. In their study, they used pain-free grip strength measurement which is, according to the literature, non-specific for TE patients and wrist extensor biomechanical parameter measurement [50].

### 4.1. Clinical Relevance

Despite a wide range of electronic dynamometer options and ongoing research, not one study has clearly indicated the most appropriate protocol for forearm muscle measurement. Our study considers different aspects of biomechanical examination for patients with pain during wrist performance and identifies the protocol most suitable for wrist examination in TE cases.

### 4.2. Limitation of the Study

The results were obtained based on a relatively small sample and thus require further, more comprehensive research in the future. Limb dominance was not taken into consideration in this study. We are aware that the isotonic protocol of forearm muscle biomechanical examination is a suggestion of wrist measurement, and it may not be suitable for all patients. The group of TE patients is only an example of a clinical group with wrist examination issues and thus requires further research in the future.

## 5. Conclusions

Isotonic wrist examination caused significantly less pain during forearm examination than the other two protocols and was well tolerated by all TE and CG participants. The short time required to complete the isotonic examination allows for a quick biomechanical assessment of muscles. The isotonic protocol is the most suitable method for wrist pathology examination as well as a biofeedback exercise performed on the Biodex System dynamometer.

## Figures and Tables

**Figure 1 medicina-60-01184-f001:**
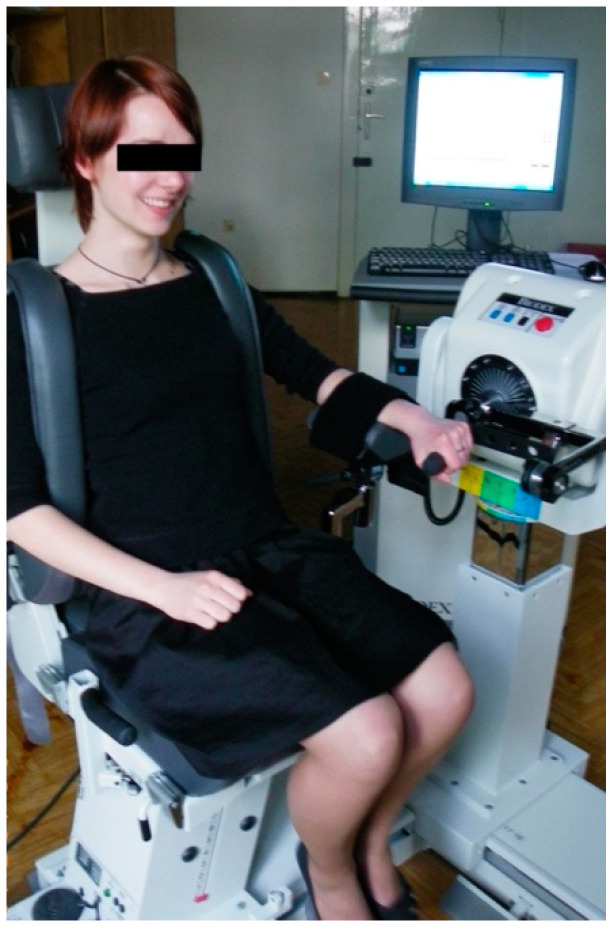
Patient position during examination.

**Table 1 medicina-60-01184-t001:** Comparison of study participant demographic data.

	PS (*n* = 10)	TE (*n* = 20)	CG (*n* = 26)	*p* (TE vs. CG)
Age				0.18 ^##^
Mean ± SD	41 ± 7	46 ± 13	39 ± 17	
Med (Q1:Q4)	36 (31–47)	48 (38.5–57)	26 (25–55)	
Height	1.65 ± 8.9	1.62 ± 10.7	1.69 ± 8.9	0.56 ^#^
Weight	69.8 ± 12.3	74.3 ± 11.2	79.7 ± 9.9	0.34 ^#^
Gender	5F/5M	13F/7M	14F/12M	−

^##^ U Mann–Whitney test, ^#^
*t* test.

**Table 2 medicina-60-01184-t002:** Protocol characteristic.

	Isokinetic (IK)	Isometric (IM)	Isotonic (IT)
I	210 deg/s	30 deg extension	0.5 Nm
II	90 deg/s	Neutral wrist position	1 Nm
III	150 deg/s	30 deg flexion	0.5 Nm
Repetitions	3	3	3

The pilot study results were not included in the main study.

**Table 3 medicina-60-01184-t003:** Pain intensity VAS (cm) during protocol examination.

		Mean ± SD	Med (Q1–Q4)	CI	*p*
Isokinetic	TE (*n* = 20)	4.7 ± 2.72	5 (3.5–6.5)	2.07–3.97	0.003 ^#^
	CG (*n* = 26)	1.48 ± 2.1	0 (0–3)	1.65–2.88	
Isometric	TE (*n* = 20)	6.55 ± 2.73	7 (4.5–9.5)	2.23–4.28	0.0005 ^#^
	CG (*n* = 26)	1.96 ± 2.24	1 (0–4)	1.77–3.08	
Isotonic	TE (*n* = 20)	2,05 ± 1.9	2 (0–4)	1.45–2.78	0.02 ^#^
	CG (*n* = 26)	0.85 ± 1.73	0–0	1.36–2.36	
TE ^ *p* < 0.00002	IK vs. IM 0.06 IK vs. IT 0.0051 IM vs. IT 0.00012				
CG ^ *p* = 0.061	NS				

^#^ *t* test, ^ one-way ANOVA, NS—non-significant.

**Table 4 medicina-60-01184-t004:** Difficulty in VAS (cm) performing protocol examinations.

		Mean ± SD	Med (Q1–Q4)	CI	*p*
Isokinetic	TE (*n* = 20)	4.35 ± 2.46	4 (2–5.5)	2.46–4.72	0.96 ^##^
	CG (*n* = 26)	4.63 ± 4.7	4 (2–7)	3.7–6.44	
Isometric	TE (*n* = 20)	7.2 ± 2.4	7 (6–9.5)	1.82–3.5	0.002 ^##^
	CG (*n* = 26)	4.48 ± 2.61	5 (2–7)	2.08–3.67	
Isotonic	TE (*n* = 20)	3.3 ± 1.53	4 (2–4.5)	1.16–2.23	0.378 ^##^
	CG (*n* = 26)	2.89 ± 2.28	2 (1–5)	1.79–3.12	
TE ^^ *p* < 0.0006	IK vs. IM 0.0019 IK vs. IT 0.00014 IM vs. IT 0.00013				
CG ^^ *p* = 0.07	NS				

^##^ U Mann–Whitney test, ^^ Kruskal–Wallis, NS—non-significant.

**Table 5 medicina-60-01184-t005:** Fatigue in VAS (cm) following examination.

		Mean ± SD	Med (Q1–Q4)	CI	*p*
Isokinetic	TE (*n* = 20)	3.9 ± 2.1	3.5 (2–5.5)	1.6–3.07	0.033 ^##^
	CG (*n* = 26)	2.67 ± 2.17	2 (2–4)	1.71–2.97	
Isometric	TE (*n* = 20)	6 ± 2.2	7 (5–7.5)	1.67–3.21	0.0001 ^##^
	CG (*n* = 26)	3.41 ± 2.08	3 (2–5)	1.64–2.85	
Isotonic	TE (*n* = 20)	1.75 ± 1.86	1 (0–3)	1.41–2.72	0.64 ^##^
	CG (*n* = 26)	1.85 ± 1.56	2 (1–3)	1.23–2.14	
TE ^ *p* < 0.00001	IK vs. IM 0.0059 IK vs. IT 0.0048 IM vs. IT 0.0012				
CG ^^ *p* = 0.02	IK vs. IM 0.35 IK vs. IT 0.28 IM vs. IT 0.013				

^##^ U Mann–Whitney test, ^ one-way ANOVA, ^^ Kruskal–Wallis.

**Table 6 medicina-60-01184-t006:** Percent of participants completing protocol examinations and personal preferences.

	Isokinetic (IK)		Isometric (IM)		Isotonic (IT)	
	TE *n* = 20	CG *n* = 26	TE *n* = 20	CG *n* = 26	TE *n* = 20	CG *n* = 26
Examination completion (%)	17 (85)	26 (100)	10 (50)	26 (100)	20 (100)	26 (100)
Preferences (%)	1 (5)	6 (23)	1 (5)	0	18 (90)	20 (77)

**Table 7 medicina-60-01184-t007:** Summary of advantages and limitations of study protocols.

	Advantages	Limitations
Isokinetic	Dynamic examination, quick biomechanical evaluation of patients’ forearm muscle condition	Low constant speed, high machine resistance Low machine resistance, high speed values (often unattainable for patients)
Isometric	Easy to perform even for patients with pain and ROM restrictions	Static measurement
Isotonic	Dynamic examination, quick biomechanical evaluation of patients’ forearm muscle condition	No muscle strength measurement

**Table 8 medicina-60-01184-t008:** Parameters available to be tested in study protocols.

	Isokinetic	Isometric	Isotonic
Peak Velocity [deg/s]	−	−	+
Peak Velocity/Body Weight [%]	−	−	+
Average Peak Velocity [deg/s]	−	−	+
Time to Peak Velocity [ms]	−	−	+
Angle of Peak Velocity [deg]	−	−	+
Velocity at 30.0° [deg/s]	−	−	+
Velocity at 0.18 sec [deg/s]	−	−	+
Peak Torque [Nm]	+	+	−
Peak Torque/Body Weight [%]	+	+	−
Average Peak Torque [Nm]	+	+	−
Acceleration Time [ms]	+	−	+
Deceleration Time [ms]	+	−	+
Relaxation Time [s]	−	+	−
Contraction Time [s]	−	+	−
Coefficiency of Variation	+	+	+
Total Work [J]	+	−	+
Maximal Repetition Total Work [J]	+	−	+
Maximal Work Repetition	−	−	+
Work/Body Weight [%]	−	−	+
Work First Third [J]	−	−	+
Work Last Third [J]	−	−	+
Work Fatigue [%]	−	−	+
Average Power [W]	+	−	+
ROM	+	−	+
Impulse [N-M]	−	+	−
Agonist/Antagonist Ratio [%]	+	+/− (only for dominant limb)	+

(+)—available, (−)—non- available.

## Data Availability

The raw data supporting the conclusions of this article will be made available by the authors on request.

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
