# Peer review of "Biomechanical Examination of Wrist Flexors and Extensors with Biodex System Dynamometer—Isometric, Isokinetic and Isotonic Protocol Options"

_medicina, 2024, doi:10.3390/medicina60071184_

Round 1

Reviewer 1 Report

Comments and Suggestions for Authors

This study is meaningful by deriving an appropriate protocol for wrist examination.

However, it seems necessary to revise the part below.

Introduction

It is not easy to understand that the introduction of this study will be conducted on Tennis elbow patients. Why this study is important to these patients should be further described in this study.

Method

- Line 93-95: The authors said that a pilot study was conducted to determine the protocol variables. It would be nice to describe what parameters were determined through the PS.

- A clear basis for sample size should be provided (ex, G-power)

- Table: In general, the * sign is used to indicate significance, so it is better to use a different symbol. It can cause confusion for readers.

Discussion

- Line 236-239: The subjects of this study are not the elderly. It is somewhat leap to say that it is a protocol for muscle aging. Therefore, additional explanation should be described.

- Line 282-286: patient's condition is too broad. Since this study was conducted only on patients with TE, it is recommended to reduce this expression. In particular, "Limitation of study" also said that it may not be suitable for all patients, so this part should be correct.

Author Response

Comment 1:

Introduction

It is not easy to understand that the introduction of this study will be conducted on Tennis elbow patients. Why this study is important to these patients should be further described in this study.

Response 1:

Thank You for Your comment. We have added in the introduction short summary why this study may be used in patients with tennis elbow diagnosis. Line 63-69

“This study may be beneficial for muscles biomechanical parameters measurement especially interconnected with muscle tendinopathies which highly influence patient mechanical abilities and decrease quality of life. One of the top examples are patients with lateral epicondyle entezopathy – tennis elbow (TE). Knowledge about wrist extensors biomechanical parameters function and potential decrease may be beneficial for patients treatment because it will provide objective measurement and provide modification of patients rehabilitation program. “

Comment 2:

- Line 93-95: The authors said that a pilot study was conducted to determine the protocol variables. It would be nice to describe what parameters were determined through the PS.

Response 2:

Thank You for the suggestion. We have accordingly to your comment added from line 142 to line 144 short information within the paper about the measured variables. Our specific analysis of each protocol variables will be published in additional scientific article with more specific biomechanical data analysis.

“The pilot study began with isometric protocol to assess the forearm muscles peak torque, according to peak torque we determined the isotonic protocol limitations. After isotonic measurement we determined the constant speed velocity for isokinetic protocol.”

Comment 3:

- A clear basis for sample size should be provided (ex, G-power)

Response 3:

Thank You for this comment. We have added short information about Gpower analysis from line 173-175.

“G*Power analysis provide 45 total sample size within 0,95 power measured with G.Power 3.1 ® software shared by Heinrich Heine Dusseldorf Univeristy (Axel Buchner, Edgar Erdfelder, Franz Faul, Albert-Georg Lang).”

Comment 4:

- Table: In general, the * sign is used to indicate significance, so it is better to use a different symbol. It can cause confusion for readers.

Response 4:

Thank You for your suggestion. We have changed the ** symbols for “# and ##” symbols. 

Comment 5:

- Line 236-239: The subjects of this study are not the elderly. It is somewhat leap to say that it is a protocol for muscle aging. Therefore, additional explanation should be described.

Response 5:

Thank You for the comment. We have decided to mention Driessche et al. research because it is presenting clear information that isotonic protocol may be used for muscles assessment despite agaging process. We added presented information within the paper. Line 259-263

“Isotonic protocol may be sufficient for muscles  assessment not only for patients in pain and but also for patients with additional muscle weakness. In our study we have focused on patients with muscle pain during performing functional tasks with resistance but according to the literature it is providing more possibilities among different group of patients. “

Comment 6:

- Line 282-286: patient's condition is too broad. Since this study was conducted only on patients with TE, it is recommended to reduce this expression. In particular, "Limitation of study" also said that it may not be suitable for all patients, so this part should be correct.

Response 6:

Thank You for Your suggestion. According to comment we have added to clinical relevance and study limitations further information.

Line 321-323

“Our study considers different aspects of biomechanical examination due for patients with pain during wrist performance and identifies the protocol most suitable for wrist examination in TE cases.”

Line 329-330

“Group of TE patients is only an example of clinical group with wrist examination issues and thus, require further research in future.”

Reviewer 2 Report

Comments and Suggestions for Authors

Thanks for the opportunity of reviewing the present manuscript. The authors presented an interesting and potentially valid work but some aspect must be better clarified. 

Here below my comments to the authors:

The tests the authors refer to cannot be categorized solely under biomechanical exams, as isometric, isokinetic, and isotonic tests aim to assess muscle contractile performance/characteristics, falling into physiological characterization rather than purely biomechanical evaluation. Biomechanical aspects are part of such evaluations but not sure is the main focus in this context. The authors may create confusion for the reader by using inappropriate terms. Authors may consider updating their work accordingly but please, taking this just as a suggestion.

Instead of specifying which population each test may be more suitable for, it is important to specify the type of information each test reveals and why, depending on the aim, different tests are important to select. So, in the introduction for example, after defining each test, it could be a good idea to explain which kind of key information each test is able to provide and why could be important to be used in different contexts and to achieve different aims.

The authors could report a table with the results of the different tests (e.g., peak torque data) to provide a clearer idea of the differences in intensity between the three protocols.

The discussion section, lines 226-228, is not clear and contains broad concepts. The authors should better specify why each test is important for assessing specific aspects. Both isometric and isotonic tests are used in sport performance evaluations, while isokinetic tests can be used in several clinical settings, including rehabilitation procedures not only for athletes.

The authors repeatedly state that the isotonic test is the best without expanding on why and without providing full insights or citing previous studies (e.g., lines 229-239).

In general, the final statements of the authors should be supported by a stronger discussion, more clearly presenting the advantages and key information obtainable through the isotonic test from the study's target population.

Author Response

Comment 1:

The tests the authors refer to cannot be categorized solely under biomechanical exams, as isometric, isokinetic, and isotonic tests aim to assess muscle contractile performance/characteristics, falling into physiological characterization rather than purely biomechanical evaluation. Biomechanical aspects are part of such evaluations but not sure is the main focus in this context. The authors may create confusion for the reader by using inappropriate terms. Authors may consider updating their work accordingly but please, taking this just as a suggestion.

Response 1:

Thank You for your suggestion. It is correct that the focus of this paper is not the biomechanical aspects of different variables in each protocol. We are planning to publish additional scientific paper with specific analysis of biomechanical parameters. The main goal of this paper was to present the possibilities of biomechanical testing of wrist extensors and flexors among patients with pain. In our experience for wrist measurement isotonic protocol is quite difficult for patients and often makes it not possible to complete. It is why we have decided to publish our first part of paper with randomized study of objective measurement of different protocols. This paper was also created mainly for clinicians who are not experts in biomechanical field and very often are looking for wrist muscles objective measurement.

Comment 2:

Type of information each test reveals and why, depending on the aim, different tests are important to select. So, in the introduction for example, after defining each test, it could be a good idea to explain which kind of key information each test is able to provide and why could be important to be used in different contexts and to achieve different aims.

Response 2:

Thank You for your comment. It is why we have implied the information within the Table 7 with basic information of advantages and disadvantages, but we have also added the info about what kind of results readers may expect due to their needs. Line 224-230

“Due to the assessed protocols isotonic protocol may be concerned as accurate for patients wrist extensors and flexors biomechanical dynamic measurement with pain issues and muscle strength decrease. Isotonic protocol may be concerned as accurate for patients wrist extensors and flexors biomechanical dynamic measurement when patient is able to perform higher restriction activities and among sportsman. Isometric protocol may be concerned as accurate for patients wrist extensors and flexors biomechanical static measurement.”

Comment 3:

The authors could report a table with the results of the different tests (e.g., peak torque data) to provide a clearer idea of the differences in intensity between the three protocols.

Response 3:

Thank You for Your suggestion. It is why we decided to present clear division between the biomechanical parameters in Table 8. We are also planning to publish another article with specific analysis of each protocol data among TE patients and control group.

Comment 4:

The discussion section, lines 226-228, is not clear and contains broad concepts. The authors should better specify why each test is important for assessing specific aspects. Both isometric and isotonic tests are used in sport performance evaluations, while isokinetic tests can be used in several clinical settings, including rehabilitation procedures not only for athletes.

In general, the final statements of the authors should be supported by a stronger discussion, more clearly presenting the advantages and key information obtainable through the isotonic test from the study's target population.

The authors repeatedly state that the isotonic test is the best without expanding on why and without providing full insights or citing previous studies (e.g., lines 229-239).

Response 4:

Thank You for Your comment. We strongly agree that it is crucial for readers to understand each protocol benefits and aspects. It is why we have presented the information within Table 7 and Table 8 and in short summary which was added to the paper after reviewer suggestion. The main challenge with wrist extensors and flexors biomechanical examination is that there are no studies which are indicating which examination is suitable for patients with pain and restricted ROM active or static examination. Our study as we mentioned is part of bigger research which we would like to publish in future with full biomechanical assessment. We have implemented part of our research in publication:

Soczka, A., Jokiel, M., Bonczar, M. et al. Biomechanical evaluation of the wrist after scaphotrapeziotrapezoid arthrodesis. Eur J Orthop Surg Traumatol (2024). https://doi.org/10.1007/s00590-024-03931-9

Due to Your suggestions, we have added to the study further information:

Line 306-317

Albanese GA et al. indicate that it is difficult to define framework for patients with forearm muscles issues that will be presented as the dynamic fatigue related changes associated with wrist function. In their work they used superficial electromyography protocol which enables the measurement of muscle fatigue but do not reveal specific muscles biomechanical parameters such as: muscle torque, velocity or muscle fatigue. It is why isotonic protocol may be considered as a usual tool for measuring patients muscle fatigue during active performance.

Hill CA et al. during their examination of patients with TE also indicated that there are biomechanical limitations for patients objective assessment. In their study they used pain-free grip strength measurement which is according to the literature non-specific for TE patients and wrist extensors biomechanical parameters biomechanical measurement.

Reviewer 3 Report

Comments and Suggestions for Authors

The article presents the study on the examination of wrist flexors and extensors using Biodex System dynamometer with implementation and further evaluation of different protocols: isometric, isokinetic and isotonic. Authors provided the basis, description and discussed the results of implementation the above protocols which allowed to determone optimal option for the wrist examination purpoces. These results could be useful in clinical practice because of considering possible wrist traumas during the examination protocol design. The article is good for publishing.

Author Response

Comment 1:

The article presents the study on the examination of wrist flexors and extensors using Biodex System dynamometer with implementation and further evaluation of different protocols: isometric, isokinetic and isotonic. Authors provided the basis, description and discussed the results of implementation the above protocols which allowed to determine optimal option for the wrist examination purposes. These results could be useful in clinical practice because of considering possible wrist traumas during the examination protocol design. The article is good for publishing.

Response 1:

We would like to thank You for your comments. We strongly believe that this work is much more to come in further research, and we would be honored to present our research results in near future.

Round 2

Reviewer 1 Report

Comments and Suggestions for Authors

The authors all modified it appropriately.

Reviewer 2 Report

Comments and Suggestions for Authors

Thank to the authors for considering my suggestions. I do not have further comments and in my opinion, the manuscript improved significantly after the peer review process.

Good luck to the authors for the next steps of the revision process.